# Accuracy of Zygomatic Implant Placement Using a Full Digital Planning and Custom-Made Bone-Supported Guide: A Retrospective Observational Cohort Study

**DOI:** 10.3390/dj11050123

**Published:** 2023-05-04

**Authors:** Francesco Gallo, Francesco Zingari, Alessandro Bolzoni, Selene Barone, Amerigo Giudice

**Affiliations:** 1Istituto Stomatologico Italiano, Via Pace, 21, 20161 Milano, Italy; 2Ospedale Galeazzi-Sant’Ambrogio, Via Belgioioso 173, 20161 Milano, Italy; 3Private Practice, Via Francesco Sforza, 35, 20123 Milano, Italy; 4Unit of Oral Surgery and Pathology, Department of Health Sciences, Magna Graecia University of Catanzaro, Viale Europa, 88100 Catanzaro, Italy

**Keywords:** zygomatic implants, accuracy, data, computer-assisted surgery, maxillary atrophy, guided zygomatic implant surgery, denture, implant-supported

## Abstract

The aim of the study was to evaluate the accuracy of zygomatic implant placement using customized bone-supported laser-sintered titanium templates. Pre-surgical computed tomography (CT) scans allowed to develop the ideal virtual planning for each patient. Direct metal laser-sintering was used to create the surgical guides for the implant placement. Post-operative CT scans were taken 6 months after surgery to assess any differences between the planned and placed zygomatic implants. Qualitative and quantitative three-dimensional analyses were performed with the software Slicer3D, recording linear and angular displacements after the surface registration of the planned and placed models of each implant. A total of 59 zygomatic implants were analyzed. Apical displacement showed a mean movement of 0.57 ± 0.49 mm on the X-axis, 1.1 ± 0.6 mm on the Y-axis, and 1.15 ± 0.69 mm on the Z-axis for the anterior implant, with a linear displacement of 0.51 ± 0.51 mm on the X-axis, 1.48 ± 0.9 mm on the Y-axis, and 1.34 ± 0.9 mm on the Z-axis for the posterior implant. The basal displacement showed a mean movement of 0.33 ± 0.25 mm on the X-axis, 0.66 ± 0.47 mm on the Y-axis, and 0.58 ± 0.4 mm on the Z-axis for the anterior implant, with a linear displacement of 0.39 ± 0.43 mm on the X-axis, 0.42 ± 0.35 mm on the Y-axis, and 0.66 ± 0.4 mm on the Z-axis for the posterior implant. The angular displacements recorded significative differences between the anterior implants (yaw: 0.56 ± 0.46°; pitch: 0.52 ± 0.45°; roll: 0.57 ± 0.44°) and posterior implants (yaw: 1.3 ± 0.8°; pitch: 1.3 ± 0.78°; roll: 1.28 ± 1.1°) (*p* < 0.05). Fully guided surgery showed good accuracy for zygomatic implant placement and it should be considered in the decision-making process.

## 1. Introduction

The generalized resorption of the alveolar process can produce severe resorption at the maxillary and mandibular bones, and in some cases, it may prevent the use of traditional implant treatment. Since 1989, zygomatic implants (ZI) have been used in severe maxillary atrophies as an alternative therapy to bone augmentation techniques. However, it is mandatory that the apex of this long implant is precisely positioned in the zygomatic bone.

As is widely known, in dental implant surgery, accurate three-dimensional positioning is essential to obtaining optimal results for a proper prosthetic rehabilitation [1]. In recent years, the combination of cone beam computed tomography (CBCT) and computer-aided design/computer-aided manufacturing (CAD/CAM) technology began to play an important role in the field of oral implantology [2]. This technology allows the production of different types of static surgical guides, using 3D printing technology [3,4]. As reported in a recent review, the accuracy of implant placement with computer-guided surgery has been reported in many studies (47) of traditional implantology [5]. Deviations between the virtually planned and the placed implants might represent an aggregate of errors, from imaging through data processing to guiding the placement during surgery [6,7]. However, different authors reported that digital surgical guides can improve the accuracy of implant placement [5,6,7,8,9].

Zygomatic implant rehabilitation is an alternative treatment for patients with severe maxillary atrophy to avoid bone-lifting or grafting procedures [10]. These long implants are inserted in a region with limited space and visibility. The implant’s apex must lie completely in the zygomatic bone both to respect many anatomical limitations and to achieve the maximum bone-implant contact [11]. Hung et al. identified the posterior superior region and the central region of the zygomatic bone as suitable areas for implant tip placement [12]. In this technique, a proper position is crucial to place the ZI without functional and aesthetic complications [13]. The accuracy of the diagnostic and planning phases and the skill and experience of the operators are key factors in this surgery [14,15,16].

To date, guided surgery for the conventional implant is widely accepted as high-precision surgery [17,18,19]. For the same reason, guided surgery for the ZI based on bone-supported drill templates appears to be useful for increasing safety and accuracy [6]. Obtaining the correct implant angulation is so crucial, especially for the multiple and contemporary ZI placements to achieve a complete restoration of the upper atrophic maxilla [20]. Discrepancies between the planned and the real implant position have become a critical point in this advanced implant surgery. Consequently, an assessment of clinical accuracy is required to determine whether guided surgery errors are clinically acceptable [21,22].

The aim of this study was to perform a three-dimensional (3D) analysis to investigate the accuracy of a novel statical surgical guide applied to zygomatic implant placement using a reliable transfer guide from a planned cooperating theatre. The primary outcome was to compare the planned and the post-operative implant positions, evaluating angular discrepancies and linear deviations in all three spatial axes. No discrepancy between the planned and placed ZIs was considered as the null hypothesis.

## 2. Materials and Methods

### 2.1. Study Design

A retrospective observational cohort study was designed, following the STROBE guidelines. The medical protocol and ethics followed the Declaration of Helsinki to promote and ensure respect and benefits for all enrolled subjects, protecting their health and rights. The Ethical committee of Central Region of Calabria approved the study (n° 252/15 July 2021).

### 2.2. Study Sample

The study sample included CBCTs of patients who underwent ZI rehabilitation, collected from July 2021 to November 2022. To use radiologic data for scientific purposes, a specific informed consent form was signed by all patients.

Selecting the database of subjects treated with dental rehabilitation supported by zygomatic implants, all patients who completed the digital protocol were considered for enrollment. The following inclusion criteria were established: (1) patients with an extreme maxillary atrophy that interfered with conventional implant placement; (2) patients treated with a full digital planning and guided surgery; (3) treatment planning with one or two (anterior implant, AI; posterior implant, PI) zygomatic implants for each side; (4) good general health.

The exclusion criteria included patients treated without pre-operative digital planning or with free-hand surgery, or patients with incomplete radiographic records.

### 2.3. Data Collection Method

All patients were submitted to pre-operative CBCT scans (T1) and post-surgical CBCT scans six months after surgery (T2) (slice thickness 0.5 mm; scan time 0.4 s; 8.0 mA; 105.0 kVc peak; 7.2 s; field of view: 15 × 13 mm). The indication for the post-operative CBCT focused on the necessity to objectively assess the post-surgical health of the maxillary sinus. To avoid errors due to an inadequate number of teeth, for some patients a stereolithographic (SLA) radiographic template with radiopaque fiducial markers was designed and fabricated according to the plate. Six fiducial markers were distributed on each side (buccal and lingual) of the radiographic template. According to the surgical objectives, pre-operative digital planning was performed for each patient using the EZplan Real Guide software (Noris Medical Ltd., Nesher, Israel) to determine the ideal position of each ZI and to design the CBCT-derived bone-assisted surgical guide (Figure 1). The EZgoma^®^ guide (Noris Medical Ltd., Nesher, Israel) was exported as a standard triangulation language (STL) file and then fabricated with the metal 3D printing process (SLM Technology Sisma MySint 100 Titanium Degree23, Sisma SRL, Italy).

### 2.4. Surgical Procedure

The same expert surgeon (FG) performed the 3D planning and the surgical treatment for all patients. After the incision of the palatal mucosa and reflection of the soft tissues up to the level of the zygoma, and after drilling through a prefabricated slide as indicated by the pre-operative virtual surgical plan, the zygomatic fixture was placed following the manufacturer’s instructions (Noris Medical Ltd., Nesher, Israel).

During installation, different drills were used, starting with a circular cutter, then switching to a 2 mm-diameter drill, continuing with a 2.9 mm drill and ending with a 3.5 mm-diameter drill that was 45 mm in length to allow the insertion of a 45 mm zygomatic implant. This specific zygomatic fixture had two diameters on the same fixture: 3.9 mm at the top and 4.5 mm at the level of the upper jaw. The surgery minimally involved the maxillary sinus. The drilling was performed with a small sinus cleft in the outer cortex of the sinus. This procedure is commonly performed for the installation of extra-sinus zygomatic implants (Figure 2A–C).

### 2.5. Comparative 3D Analysis

The resulting STL files of the planned ZI were exported for the analysis. The post-operative CBCT was processed, and a tissue density segmentation was performed to isolate the ZI from the surrounding bone using the ITK-SNAP software (version 3.8.0; http://www.itksnap.org).

The morphometric analysis was conducted on the 3DSlicer software (version 4.13.0; http://www.slicer.org) to compare digitally planned and post-surgical 3D reconstructions of the ZI. Using the “Model Maker” tool, the 3D surface models of each segmentation were developed. Using the “EasyClip” tool, the models of the planned and placed ZIs were cut, dividing the right and left side. The surface registration of the planned and placed ZI models was performed to superimpose the models before the following analyses.

### 2.6. Qualitative Analysis

Using the “Model-to-Model Distance” and “Shape Population Viewer” tools, colormaps were created to visualize any displacement between the digitally planned and T2 models. The absence of clinical surgical displacement (0 to 1 mm) is indicated by green.

### 2.7. Quantitative Analysis

Quantitative analysis was performed by an independent operator (SB). The examiner repeated the measurements after 2 weeks, and an intra-reliability value of 0.93 was recorded. The “Mesh Statistics” tool was used to quantify the mean difference between the surface meshes of the digitally planned and T2 models. Additionally, landmark-based quantitative assessments were obtained by the “Q3DC” tool. Linear deviations (in millimeters, mm) between digitally planned and T2 models were calculated in the three spatial axes, placing two points on the 3D models of each implant: the center of the apical surface (A) and the center of the basal surface (B). Angular deviations (degrees, °) between the long axis of digitally planned and T2 models were recorded.

### 2.8. Study Variables and Outcomes

The primary outcome variable was the displacement between the planned and post-surgical ZI models. The following target deviations were defined and calculated (Figure 3):
-An operator-independent calculation recorded the mean displacement between the planned and T2 ZIs comparing the 3D surface meshes.-Linear differences at the implant’s apex and base were recorded in anteroposterior (X-axis), upper–lower (Y-axis), and medio-lateral (Z-axis) directions in mm. -The angular deviation between the planned and T2 ZIs was determined, calculating yaw, pitch, and roll of the long axis of each implant (°).

Other study variables recorded: patients’ age and gender, and number of placed implants.

### 2.9. Statistical Analysis

The database was created using a dedicated Excel file (Microsoft, Redmond, WA, USA). Statistical analysis was performed using the software STATA (STATA 11, StataCorp, College Station, TX, USA).

Descriptive statistics recorded mean and standard deviation for continuous quantitative variables, absolute and relative frequencies for categorical data. Box plots were used to estimate data outliers.

To compare the planned and post-surgical implants, the analysis of variance was performed, using the two-tailed Student *t*-test for normal distributions and Wilcoxon test for asymmetrical distributions. The Shapiro–Wilk test allowed an evaluation of the type of distribution for each variable. The level of significance was set at α = 0.05.

A power analysis was finally performed to guarantee at least a level of 80% (effect size 0.3; α = 0.05; sample size = 59).

## 3. Results

Nineteen patients were included in the study and a total of 59 implants were examined. Details of the study sample are reported in Table 1. 

In all cases, implant placement was performed using a bone-supported surgical guide that showed a stable fit with no need of bone adjustments. To fix the surgical guide, metal screws were inserted in all templates. No lesion of the surgical template nor bone fractures occurred during the surgeries. 

All patients were rehabilitated with almost one zygomatic implant placement. Most of them received four ZIs (63.2%). No implants were lost, indicating a survival rate of 100% at 6 months follow-up. 

### 3.1. Qualitative Analysis

Qualitative analysis recorded semitransparent overlays of the placed and planned zygomatic implants (Figure 4).

### 3.2. Quantitative Analysis

Descriptive statistics of linear implant displacements are reported in Table 2.

The surface displacement at T2 compared to the planned model showed a mean difference of 0.26 ± 0.12 mm on the right side and 0.22 ± 0.15 mm on the left side (*p* = 0.16).

According to the different coordinates, on X-axis the apical displacement showed a mean movement of 0.57 ± 0.49 mm and 0.51 ± 0.51 mm for the AI and PI, respectively (*p* > 0.05). The upper–lower component (Y-axis) showed an upper position of the AI at T2 with a mean discrepancy of 1.1 ± 0.6 mm, while a lower position (1.48 ± 0.9 mm) was recorded for the Pis (*p* > 0.05). On the Z-axis, a lateral displacement occurred at T2 compared to the planned models, both for the Ais (1.15 ± 0.69 mm) and Pis (1.34 ± 0.9 mm) (*p* > 0.05).

Basal displacement of the Ais and Pis showed a mean anterior displacement on the X-axis of 0.33 ± 0.25 mm and of 0.39 ± 0.43 mm, respectively (*p* > 0.05). Comparing the T2 models with respect to the planned models in the upper–lower component (Y-axis), a significative difference between the Ais (0.66 ± 0.47 mm) and Pis (0.42 ± 0.35 mm) was recorded (*p* = 0.037). On the Z-axis, a lateral displacement occurred both for the Ais (0.58 ± 0.4 mm) and Pis (0.66 ± 0.4 mm) (*p* > 0.05).

Table 3 reports the angular displacement of implant orientation between the planned and placed models. A significative difference was recorded comparing the mean discrepancy between the Ais (yaw: 0.56 ± 0.46°; pitch: 0.52 ± 0.45°; roll: 0.57 ± 0.44°) and Pis (yaw: 1.3 ± 0.8°; pitch: 1.3 ± 0.78°; roll: 1.28 ± 1.1°) in all directions of rotation (*p* < 0.05).

## 4. Discussion

The purpose of this study was to evaluate the accuracy of a guided surgical protocol for ZI placement by analyzing data obtained from the superimposition between the pre-operative digital planning and the post-operative CBCT scan of the treated patients. Currently, many authors consider guided surgery to increase surgical accuracy and reduce the risks of implant placement [3,5]. The primary goal is to minimize neurological issues and preserve critical anatomical features such as the orbit. To date, one of the most used procedures in ZI placement is the extra-sinus insertion, which minimizes the involvement of the maxillary sinus respiratory space and eliminates the need for membrane elevation. Additionally, thanks to this approach, the implant head is positioned at or near the top of the remaining crest, in a more favorable prosthetic position [14,15]. Pre-operative radiological measurements between anatomical landmarks can be inaccurate, posing a danger, particularly in the case of blind surgery [15]. In order to increase the accuracy in the comparison between the virtual planning with the post-operative outcomes, a 3D imaging analysis was implemented. Post-surgical CBCT allowed for a follow-up of the post-operative health of the maxillary sinus, also representing the best radiological exam to assess the implant position [16]. To our knowledge, this is one of the first in vivo studies analyzing the accuracy of ZI placement through bone-supported templates with a 3D assessment. 

Our study included 59 ZIs, most of them in a quad approach with the guidance of a bone-supported guide. As already specified, in this technique, a bone-supported surgical guide must be placed after reflecting a full-thickness flap. Compared to non-guided surgery, in the case of the ZI procedure, there is no need for more invasive maneuvers to be placed in the surgical guide. Indeed, the presence of fixation screws may help in the precision and stabilization of the template; however, due to the need for a wide normal exposure of the zygoma, the procedure may be comparable to traditional non-guided surgery. 

Following the initial hypothesis, the results of this study confirmed a negligible difference between the virtually planned and the post-operative placed implants. The primary findings emphasized the accuracy of this bone support device consisting of a single sintered titanium template placed during all the surgical procedures. The qualitative and quantitative outcomes showed a significant overlapping of the post-operative implants compared to the planned ones. The surface displacement recorded a mean difference of 0.26 ± 0.12 mm on the right-side implants and 0.22 ± 0.15 mm on the left side. Analyzing the quantitative discrepancies on the three spatial axes, the mean lateral deviation was under 0.50 mm at the apex and the base of the anterior implants. Although the posterior implants showed a slightly higher mean lateral displacement, this data, even if not significant, could be due to the greater difficulty in inserting the posterior implant in the limited space and mouth opening. The data recorded in the study showed very close linear distance values between the guided zygomatic surgery and the standard guided implant surgery.

Analyzing the accuracy of the virtual surgical planning in zygomatic implant insertion, Xing Gao, B. et al. reported a significant difference between the planned and the final implant position with a free-hand traditional surgery, especially in the angular position [23]. The study demonstrated how the transfer error from the pre-operative planning to the surgical field is a critical factor and surgical experience is still mandatory. In traditional implant surgery, the acceptable transfer error ranges from 0.3 to 0.6 mm. Naitoh et al. found that transferring the VPS to surgery using conventional teeth-supported guides resulted in an angular deviation between the planned and real position ranging from 0.5° to 14° with an average of 5.0° [24].

In the case of ZI surgery, the literature described different results, probably due to the complexity of the procedure and the length of the implants. The mean difference reported by Van Steenberghe et al. was 2.0–2.5 mm for linear discrepancies and 3 degrees for angular displacements [6]. In the systematic review by Van Assche and colleagues, the mean deviations at entry, at the apex and the angular deviation were 0.73 mm ± 0.16, 0.98 mm ± 0.20 and 3.08° ± 0.37, respectively [25].

More recently, Grecchi et al. published a cadaver study testing a titanium laser-sintered bone-supported guide in zygomatic and pterygoid implant surgery. The authors tested its accuracy on a total of 40 zygomatic and 20 pterygoid implants. They reported that the mean deviations between the planned and the placed zygomatic and pterygoid implants were, respectively: 1.69 ± 1.12 mm and 4.15 ± 3.53° for the angular deviation. Linear distance deviations: 0.93 mm ± 1.23 mm and 1.35 mm ± 1.45 mm at the platform depth, 1.35 mm ± 0.78 mm and 1.81 mm ± 1.47 mm at the apical plane, and 1.07 mm ± 1.47 mm and 1.22 mm ± 1.44 mm for the apical depth [26].

The results obtained in this study are comparable with reports of traditional tooth or bone-supported guided surgery. In the case of ZI rehabilitation, few effective methods have been reported to transfer virtual planning to the surgical field. For this reason, the surgery is mostly performed conventionally, causing some discrepancies between the virtually planned and final position of the implant. Static or navigation-guided surgery may be an efficient tool to accurately transfer VSP to the surgical field.

Concerning surgical insertion guides, several studies have described different materials to produce this intraoperative device, including silicone, synthetic resin, thermoforming films, or titanium [27]. According to their elastic characteristics, the results might change, adding errors to the attained insertion depth or angulation. Tatakis et al. explored another possible cause of mistakes that produce inaccuracies in the final implant location [28]. They suggested that one potential reason might be the space that exists between the system’s drill and the guiding unit of the insertion guides [28]. Their research revealed that differences between the inner and outer diameters of the blade and the guide sleeve have a direct impact on the accuracy [28]. Laederach et al. pointed out that a small gap cannot be avoided since significant mechanical frictional forces can develop [28]. The polymeric guides are usually easier to fabricate but with the disadvantage of being larger and sometimes having a mixed anchorage. On the contrary, the titanium guide is smaller and with a bone screw anchorage can precisely fit the planned position. Unlike other guides, zygomatic implant placement can also be completed without removing it, so the final exit point and angulation control are fully provided by the device [27,28]. Moreover, the proposed bone-supported titanium guide can offer the possibility to plan a guided conventional implant placement in the same guide. The results showed how an accurate definition of the implant starting point, trajectory, and exit point was achieved and ZI placement was associated with a more predictable implant position. 

There are some limitations that should be considered. First, a relevant impact of surgical experience on the accuracy of implant placement is to be expected, mainly for the adequate placement of the bone-supported surgical guide. In this study, all patients had been treated by one experienced clinician. Second, this study may have been underpowered by the lack of a control group treated with free-hand surgery. Third, image error, technological error, registration, calibration error, and human error are all factors that can affect guided implant surgery [29]. The imaging model and associated imaging parameters have a significant impact on image quality. The CBCT, rather than CT, may influence the accuracy of planning ZI surgery [29]. The pixel size and slice thickness may influence the planning-related software associated with the printing algorithm within and may generate a technical error. Image importation, image reconstruction, occlusion plane definition, arch curve drawing of the prosthesis-driven implant planning, and fine implant adjustment were performed with different types of pre-operative planning software. Moreover, the use of long drills instead of short traditional ones may lead to a minimal loss of precision. However, this effect should be minimized by the titanium sintered guide compared to free-hand surgery, due to guide’s specific design. In the same way, the guide may reduce the effect of a surgeon in an uncomfortable position handling rotary instruments.

## 5. Conclusions

The results obtained with this surgical procedure appear to be promising. The findings of this study may help clinicians in selecting a more accurate technique for placing ZIs. This study shows that with a bone-supported guide, the surgery may achieve a high level of accuracy even in fully edentulous patients. Despite the inherent difficulties of osteotomy driving the angle formation due to the length of the zygomatic implant, the guided ZI surgery showed a minimal difference between the planned and positioned implants. Additional studies and randomized clinical trials comparing guided versus free-hand surgery are required to assess the predictability of this procedure. As a result, when the zygomatic implant is necessary, fully guided surgery should be considered in the decision-making process for the surgical approach.

## Figures and Tables

**Figure 1 dentistry-11-00123-f001:**
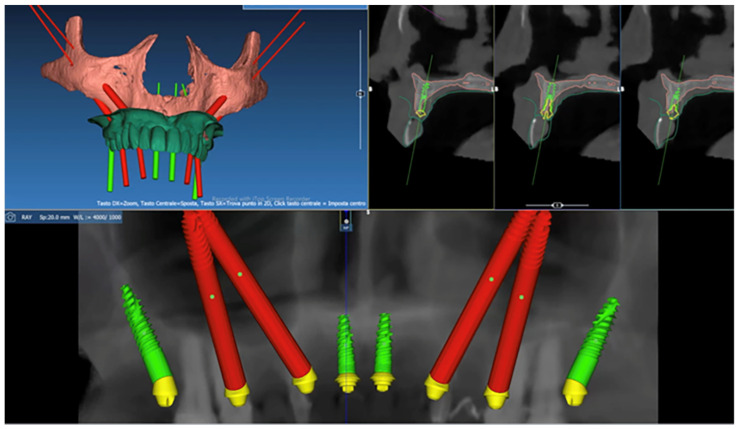
Pre-operative digital planning using the EZplan Real Guide software (Noris Medical Ltd., Nesher, Israel).

**Figure 2 dentistry-11-00123-f002:**
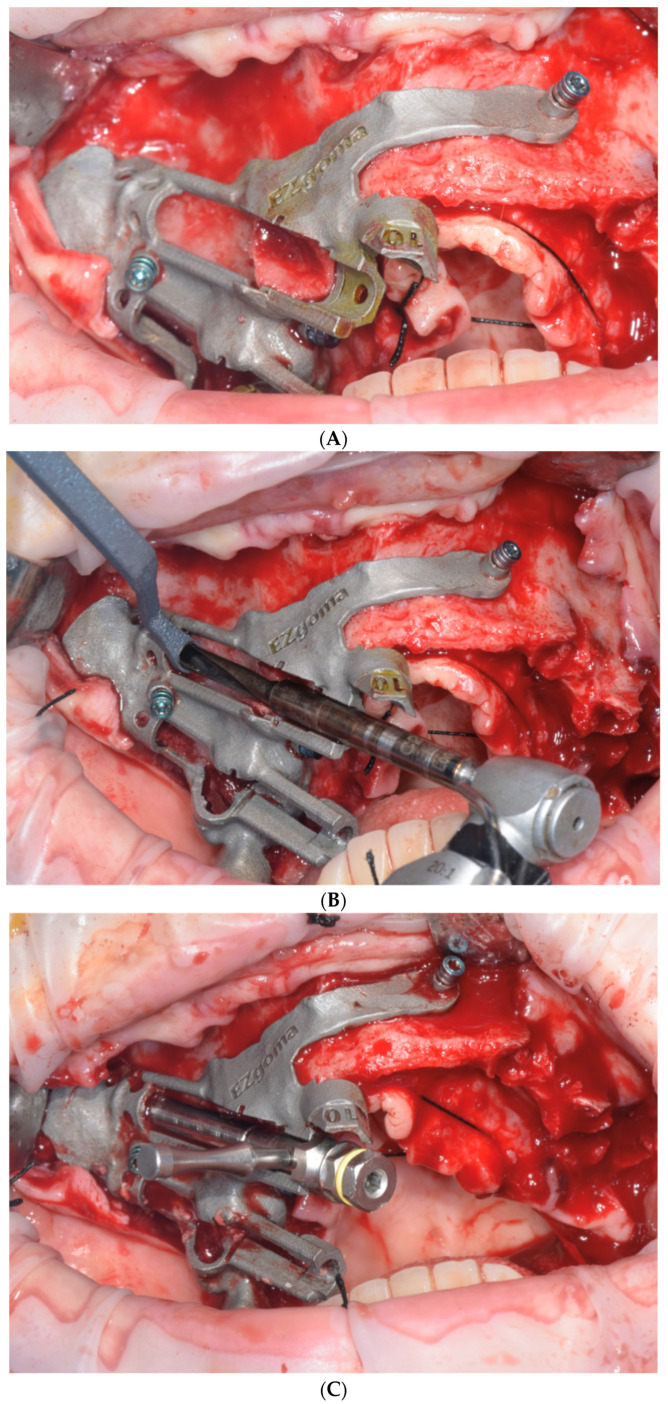
(**A**) Full thickness flap and fixation of EZgoma^®^ guide; (**B**) Implant site preparation with dedicated drills for the zygomatic fixture; (**C**) The zygomatic implant was screwed in place by a dedicated mounter.

**Figure 3 dentistry-11-00123-f003:**
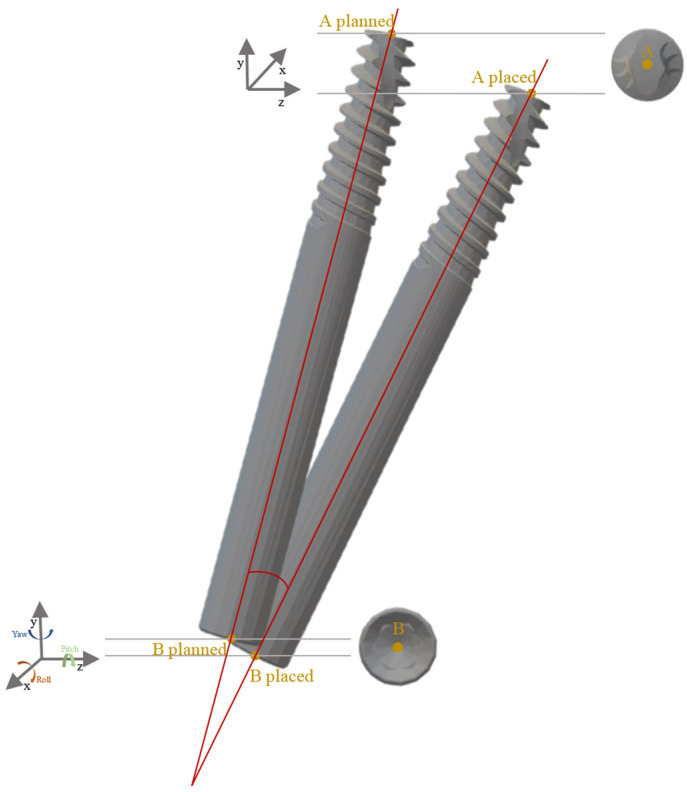
Measurement of deviation between planned and placed implants.

**Figure 4 dentistry-11-00123-f004:**
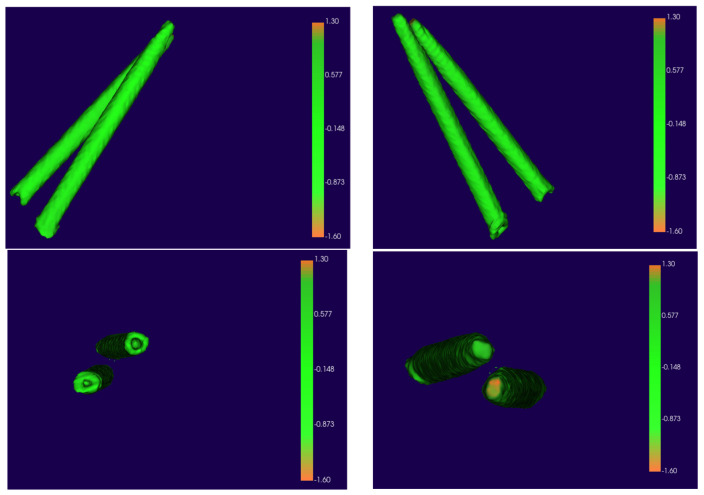
Automated colormaps generated by the software. Different visualizations were recorded: anterior view; posterior view; implant base; implant apex.

**Table 1 dentistry-11-00123-t001:** Descriptive statistics of the study sample.

Demographic Variables	Study Sample
Patients	19
Implants	59
Sex	
Female (%)	11 (57.9)
Age (years)	61 ± 3
Number of placed implants for each patient (%)	
2	6 (31.5)
3	1 (5.3)
4	12 (63.2)

**Table 2 dentistry-11-00123-t002:** Linear measurements of planned and placed implants.

	Min	Q1	Q2	Q3	Max	Mean	SD
**Surface displacement**							
Right	0.042	0.2	0.26	0.31	0.48	0.26	0.12
Left	0.014	0.15	0.23	0.3	0.49	0.22	0.15
**X-axis**							
A_R_AI	0.03	0.25	0.41	0.58	1.96	0.52	0.51
A_R_PI	0.013	0.21	0.37	0.58	2.5	0.58	0.7
B_R_AI	0.01	0.06	0.3	0.57	0.81	0.34	0.26
B_R_PI	0.04	0.09	0.22	0.52	0.63	0.31	0.23
A_L_AI	0.06	0.18	0.63	0.84	1.57	0.63	0.48
A_L_PI	0.14	0.24	0.39	0.59	1.04	0.45	0.27
B_L_AI	0.008	0.15	0.21	0.45	0.9	0.32	0.25
B_L_PI	0.07	0.17	0.29	0.47	2.1	0.47	0.56
**Y-axis**							
A_R_AI	0.21	0.54	1.04	1.68	2.2	1.12	0.63
A_R_PI	0.63	1.26	1.9	2.04	3.35	1.78	0.8
B_R_AI	0.006	0.2	0.46	0.75	1.7	0.54	0.46
B_R_PI	0.06	0.16	0.43	0.68	1.2	0.47	0.38
A_L_AI	0.27	0.58	0.97	1.45	2.34	1.04	0.57
A_L_PI	0.07	0.73	0.84	1.64	3	1.2	0.95
B_L_AI	0.036	0.44	0.77	0.94	1.69	0.78	0.48
B_L_PI	0.08	0.12	0.29	0.5	1.05	0.38	0.34
**Z-axis**							
A_R_AI	0.02	0.65	1.05	1.63	2.6	1.14	0.76
A_R_PI	0.32	0.9	1.55	1.98	4.1	1.63	1.12
B_R_AI	0.02	0.26	0.5	0.57	1.09	0.48	0.32
B_R_PI	0.23	0.38	0.62	0.9	1.5	0.72	0.43
A_L_AI	0.1	0.7	0.99	1.6	2.7	1.17	0.65
A_L_PI	0.14	0.7	0.93	1.36	2.36	1.07	0.6
B_L_AI	0.02	0.36	0.7	0.89	1.89	0.68	0.45
B_L_PI	0.08	0.23	0.66	0.8	1.22	0.62	0.38
**3D distance**							
A_R_AI	0.46	1.14	1.65	2.3	3.66	1.78	0.92
A_R_PI	0.96	2.16	2.44	3.28	4.62	2.67	1.16
B_R_AI	0.23	0.6	0.79	1.13	2.03	0.89	0.47
B_R_PI	0.3	0.62	0.8	1.25	1.98	0.97	0.5
A_L_AI	0.56	1.2	1.56	2.2	3.83	1.75	0.87
A_L_PI	1	1.2	1.48	2.4	3.6	1.84	0.85
B_L_AI	0.31	0.88	1.03	1.3	2.65	1.15	0.59
B_L_PI	0.1	0.57	0.8	1.14	2.4	0.95	0.63

R = right; L = left; A = apical; B = base; AI = anterior implant; PI = posterior implant; Q1 = first quartile, 25th percentile; Q2 = second quartile, 50th percentile; Q3 = third quartile, 75th percentile.

**Table 3 dentistry-11-00123-t003:** Angular measurements of planned and placed implants.

	Min	Q1	Q2	Q3	Max	Mean	SD
**Yaw**							
R_AI	0.12	0.27	0.33	0.74	1.76	0.56	0.45
R_PI	0.62	1.1	1.28	1.8	2.58	1.45	0.6
L_AI	0.12	0.23	0.4	0.8	1.9	0.56	0.49
L_PI	0.2	0.42	0.96	1.95	2.83	1.22	0.98
**Pitch**							
R_AI	0.03	0.19	0.21	0.46	1.99	0.43	0.5
R_PI	0.12	0.64	1.04	1.73	2.24	1.18	0.7
L_AI	0.12	0.36	0.5	0.83	1.45	0.61	0.4
L_PI	0.27	0.89	1.4	2.35	2.65	1.49	0.84
**Roll**							
R_AI	0.02	0.16	0.35	0.78	1.46	0.53	0.49
R_PI	0.08	0.5	0.83	2.67	3.28	1.41	1.23
L_AI	0.16	0.37	0.44	0.64	1.6	0.6	0.4
L_PI	0.06	0.68	1.02	1.3	4.18	1.16	1.07

R = right; L = left; AI = anterior implant; PI = posterior implant; Q1 = first quartile, 25th percentile; Q2 = second quartile, 50th percentile; Q3 = third quartile, 75th percentile.

## Data Availability

The data presented in this study are available on request from the corresponding author.

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
