# Peer review of "Accuracy of Zygomatic Implant Placement Using a Full Digital Planning and Custom-Made Bone-Supported Guide: A Retrospective Observational Cohort Study"

_dentistry, 2023, doi:10.3390/dj11050123_

Round 1
Reviewer 1 Report
The article "Accuracy of Zygomatic Implant Placement Using a Full Digital Planning and custom-made bone-supported guide" had as goal to evaluate the precision of implant placement for the zygomatic bones utilizing specially constructed bone-supported laser sintered titanium templates.
It is an interesting clinical retrospective study using a metallic guide.
- Why did the authors not consider using a polymeric guide (less cost and easy to make)? Include a discussion about it please, justifying the preference to use it.
- The abstract can be better worked to present clearly the data.
- I considered the introduction and the results enough.
Outstanding study. Congratulations
Author Response
Thank you very much for the consideration given to our paper. Thank you for your suggestions aiming to improve the research. We followed your comments and reported in the revised manuscript all the modifications in red.
The article "Accuracy of Zygomatic Implant Placement Using a Full Digital Planning and custom-made bone-supported guide" had as goal to evaluate the precision of implant placement for the zygomatic bones utilizing specially constructed bone-supported laser sintered titanium templates.
It is an interesting clinical retrospective study using a metallic guide.
- Why did the authors not consider using a polymeric guide (less cost and easy to make)? Include a discussion about it please, justifying the preference to use it.
Authors’ response: Done
- The abstract can be better worked to present clearly the data.
Authors’ response: Done.
- I considered the introduction and the results enough.
Outstanding study. Congratulations

Reviewer 2 Report
It was evaluated the article titled "Accuracy of Zygomatic Implant Placement Using a Full Digital Planning and custom-made bone-supported guide", which enrolled 19 patients (with 59 zygomatic implants).
The aim was to evaluate the precision of implant placement for the zygomatic bones utilizing specially constructed bone-supported laser sintered titanium templates.
I really appreciate the study and I support the following observation "this is one of the first in vivo studies analyzing the accuracy of ZI placement through bone-supported templates with a 3D assessment" - Congrats!
Some questions/concerns:
ABSTRACT: improve the details of the results
INTRO: well done.
M&M: it was used STROBE (great) and I suggest including more info for the Declaration of Helsinki.
It was used CT before and after surgeries - great!
Results: well done
DISCUSSION: "accuracy of a surgical-guided protocol" - why do the authors only tested metallic guide? please, insert discussion about it citing the polymeric guide as possible option, presenting pros and cons.
Author Response
Thank you very much for the consideration given to our paper. Thank you for your suggestions aiming to improve the research. We followed your comments and reported in the revised manuscript all the modifications in red.
It was evaluated the article titled "Accuracy of Zygomatic Implant Placement Using a Full Digital Planning and custom-made bone-supported guide", which enrolled 19 patients (with 59 zygomatic implants).
The aim was to evaluate the precision of implant placement for the zygomatic bones utilizing specially constructed bone-supported laser sintered titanium templates.
I really appreciate the study and I support the following observation "this is one of the first in vivo studies analyzing the accuracy of ZI placement through bone-supported templates with a 3D assessment" - Congrats!
Some questions/concerns:
ABSTRACT: improve the details of the results
Authors’ response: Done.
INTRO: well done.
M&M: it was used STROBE (great) and I suggest including more info for the Declaration of Helsinki.
It was used CT before and after surgeries - great!
Results: well done
DISCUSSION: "accuracy of a surgical-guided protocol" - why do the authors only tested metallic guide? please, insert discussion about it citing the polymeric guide as possible option, presenting pros and cons.
Authors’ response: Done

Reviewer 3 Report
The article presents an assessment of the digital protocol accuracy for planning operations of Zygomatic Implant Placement, which also includes the production of individual templates by three-dimensional printing, which is quite interesting and relevant. To achieve this goal, the authors chose a study design based on a comparative analysis of the spatial parameters of the planned (virtual) implants position and the actual result after 6 months. The purpose of the presented research is important for two reasons: 1) lack of space for implants placement, as well as the location of anatomical structures, which increases the risk of iatrogenic complications; 2) the presence of digital technologies in the protocol of planning and treatment, including additive technologies – the study of which is still active.
The article is well structured and its narration is accessible. The described preliminary (checking for the normality of the distribution) and basic (parametric and nonparametric criteria) statistical methods of analysis are correct and correspond to the purpose and hypothesis of the study. The information described in the "Materials and Methods" section is detailed enough to repeat the experiment.
The presented work leaves a positive impression, however, there are some suggestions:
1. Include the type of research in the title of the article;
2. Adjust keywords according to the database https://meshb.nlm.nih.gov/;
3. Add a column with p-value to Tables 2 and 3.
Author Response
Thank you very much for the consideration given to our paper. Thank you for your suggestions aiming to improve the research. We followed your comments and reported in the revised manuscript all the modifications in red.
The article presents an assessment of the digital protocol accuracy for planning operations of Zygomatic Implant Placement, which also includes the production of individual templates by three-dimensional printing, which is quite interesting and relevant. To achieve this goal, the authors chose a study design based on a comparative analysis of the spatial parameters of the planned (virtual) implants position and the actual result after 6 months. The purpose of the presented research is important for two reasons: 1) lack of space for implants placement, as well as the location of anatomical structures, which increases the risk of iatrogenic complications; 2) the presence of digital technologies in the protocol of planning and treatment, including additive technologies – the study of which is still active.
The article is well structured and its narration is accessible. The described preliminary (checking for the normality of the distribution) and basic (parametric and nonparametric criteria) statistical methods of analysis are correct and correspond to the purpose and hypothesis of the study. The information described in the "Materials and Methods" section is detailed enough to repeat the experiment.
The presented work leaves a positive impression, however, there are some suggestions:
- Include the type of research in the title of the article;
Authors’ response: Done.
- Adjust keywords according to the database https://meshb.nlm.nih.gov/;
Authors’ response: Done
- Add a column with p-value to Tables 2 and 3.
Authors’ response: Thanks to the reviewer for the opportunity to clarify this aspect.
Table 2 and 3 reported the linear and angular displacements between the planned and placed zygomatic implants after the landmarks identification. This data is already achieved as a difference in the three spatial axes for linear displacements and in the three rotation’s directions for angular displacements, so, unfortunately, only descriptive statistics could be done. The comparisons reported in the text are referred to the assessment of differences between the anterior and posterior implants. Considering that in these Tables we also recorded the values of the subgroups (right vs left; anterior vs posterior; apex vs base), it’s complicated to specify the p-value avoiding misunderstandings.

Reviewer 4 Report
Overall, this is a very well planned and written study. The results are interesting.
Abstract: The abstract is well written and easy to understand.
Keywords: simple for readers to understand and are relevant to the content.
Introduction: The introduction provides a good, generalized background of the topic. However, to make the introduction more substantiated, the authors could make the following improvements:
It reads “The accuracy of implant placement with computer-guided surgery has been reported in many 47 studies of traditional implantology [5].”
Please add that it is based on a systematic literature review or add more references.
Material and Methods:
This study has considered every important point required. Ethical consideration and inclusion-exclusion criteria have been explained properly.
However, the authors should explain in more detail the indications for which postsurgical CBCT scans was made. A reference justifying the need for such a CBCT should be given. Otherwise, the ethical conditions of the research could be questioned.
However, authors must describe in more detail how patients were recruited for the study-clinic, city, etc.
Repeated measurements by the same observer (SB) should be performed to ensure repeatability of measurements.
Please provide a power analysis for your final sample size.
Results: well-written results and tables are self-explanatory. Data are well presented, and no need for any supplementary figures or tables.
However, The meaning of Q1; Q2; Q3 should be indicated at table 2 and 3.
It reads ”The surface displacement at T2 compared to the planned model showed a mean dif- 240 ference of 0.26±0.12 mm on the right side, and 0.22±0.15 mm on the left side.”
Please, add p values.
Discussion: The discussion is well written.
Conclusions: Conclusions are written comprehensibly and according to the results.
Author Response
Reviewer 4
Overall, this is a very well planned and written study. The results are interesting.
Abstract: The abstract is well written and easy to understand.
Keywords: simple for readers to understand and are relevant to the content.
Introduction: The introduction provides a good, generalized background of the topic. However, to make the introduction more substantiated, the authors could make the following improvements:
It reads “The accuracy of implant placement with computer-guided surgery has been reported in many 47 studies of traditional implantology [5].”
Please add that it is based on a systematic literature review or add more references.
Authors’ response: Thank you for the suggestion. We modified. As reported in a recent review, the accuracy of implant placement with computer-guided surgery has been reported in many studies (47) of traditional implantology [5].
Material and Methods:
This study has considered every important point required. Ethical consideration and inclusion-exclusion criteria have been explained properly.
However, the authors should explain in more detail the indications for which postsurgical CBCT scans was made. A reference justifying the need for such a CBCT should be given. Otherwise, the ethical conditions of the research could be questioned.
Authors’ response: In order to determine the localization of the fixtures, a post-surgical CBCT was performed in all patients after six months. A second CBCT scan was chosen above alternative non-radiography approaches (digital impression via scan abutment) because the protocol included patients who were totally edentulous. This protocol could guarantee the most accurate comparison of the virtual planning with the postoperative data. On the other hand, the second CBCT was focused represent an available method to follow up maxillary sinus health. As reported in the paragraph, the Ethical Committee gave its approval to this matter. We described this indication in the manuscript.
Methods section: “The indication for the post-operative CBCT focused on the necessity to objectively assess the post-surgical health of the maxillary sinus.”
Discussion section: “In order to increase the accuracy in the comparison between the virtual planning with the postoperative outcomes, a 3D imaging analysis was implemented. Post-surgical CBCT allowed to follow up the post-operative health of the maxillary sinus, representing also the best radiological exam to assess the implant position [16].”
However, authors must describe in more detail how patients were recruited for the study-clinic, city, etc.
Authors’ response: Thank you for suggestion. We provided to add more informations in the methods section.
Repeated measurements by the same observer (SB) should be performed to ensure repeatability of measurements.
Authors’ response: Thank you for the recommendation. As added in the text, “The examiner repeated the measurements after 2 weeks, and an intra-reliability value of 0.91 was recorded.”
Please provide a power analysis for your final sample size.
Authors’ response: Thank you for the suggestion. However, we added the methodological procedure in the methods section. A power analysis was finally performed to guarantee at least a level of 80% (effect size 0.3; α= 0.05; sample size=59).
Results: well-written results and tables are self-explanatory. Data are well presented, and no need for any supplementary figures or tables.
However, The meaning of Q1; Q2; Q3 should be indicated at table 2 and 3.
Authors’ response: Thank you for the recommendation. We added the explanation for the tables. Q1=first quartile, 25th percentile; Q2=second quartile, 50th percentile; Q3=third quartile, 75th percentile
It reads ”The surface displacement at T2 compared to the planned model showed a mean dif- 240 ference of 0.26±0.12 mm on the right side, and 0.22±0.15 mm on the left side.”
Please, add p values.
Authors’ response: We added the p-value to record the difference. p=0.16
Discussion: The discussion is well written.
Conclusions: Conclusions are written comprehensibly and according to the results.

Round 2
Reviewer 4 Report
Accept in present form